# The global climatology of the intensity of the ionospheric sporadic $E$ layer

Bingkun Yu[1,2,3], Xianghui Xue[1,2,4], Xin'an Yue[5], Chengyun Yang[1,2], Chao Yu[1,2], Xiankang Dou[1,2], Baiqi Ning[5], and Lianhuan Hu[5]

[1]CAS Key Laboratory of Geospace Environment, Department of Geophysics and Planetary Sciences, University of Science and Technology of China, Hefei, China
[2]Mengcheng National Geophysical Observatory, School of Earth and Space Sciences, University of Science and Technology of China, Hefei, China
[3]Department of Meteorology, University of Reading, Berkshire, UK
[4]Synergetic Innovation Center of Quantum Information and Quantum Physics, University of Science and Technology of China, Hefei, China
[5]Key Laboratory of Earth and Planetary Physics, Institute of Geology and Geophysics, Chinese Academy of Sciences, Beijing, China

**Correspondence:** Xianghui Xue (xuexh@ustc.edu.cn)

**Abstract.** On the basis of S4max data retrieved from COSMIC GPS radio occultation measurements, the long-term climatology of the intensity of $E_s$ layers is investigated for the period from December 2006 to January 2014. Global maps of $E_s$ intensity show the high-spatial-resolution geographical distribution and strong seasonal dependence of $E_s$ layers. The maximum intensity of $E_s$ occurs over the midlatitudes, and its value in summer is 2–3 times larger than that in winter. A relatively

5    strong $E_s$ layer is observed at the North and South Poles, with a distinct boundary dividing the midlatitudes and high latitudes along the 60°–80° geomagnetic latitude band. The simulation results show that the convergence of vertical ion velocity could partially explain the seasonal dependence of $E_s$ intensity. Furthermore, some disagreements between the distributions of the calculated divergence of vertical ion velocity and the observed $E_s$ intensity indicate that other processes, such as the vertical motions of gravity waves, magnetic field effects, meteoric mass influx into Earth's atmosphere and the chemical processes of

10    metallic ions, should also be considered, as they may also play an important role in the spatial and seasonal variations in $E_s$ layers.

## 1 Introduction

Ionospheric sporadic $E$ ($E_s$) layers are thin-layered structures with intense, high electron densities at 90–130 km altitudes. Rocket-borne mass spectrometric measurements have shown that the $E_s$ layer mostly result from the ionization of metal atoms,

15    such as $Fe^+$, $Mg^+$, and $Na^+$ (Kopp, 1997; Grebowsky and Aikin, 2002). The $E_s$ layer mainly resides over midlatitudes and is relatively absent at the geomagnetic equator and high latitudes (Whitehead, 1989). It is widely accepted that the mechanism responsible for the $E_s$ layer formation at midlatitudes is the wind shear theory, in which the zonal and meridional winds provide vertical wind shear convergence nodes. As a result, long-lived metallic ions are forced to converge towards the wind

shear null to form a thin layer of intense metallic ionization(Whitehead, 1961; Macleod, 1966; Whitehead, 1970; Nygren et al., 1984; Whitehead, 1989; Haldoupis, 2012). In the equatorial region, the physical process of $E_s$ irregularities is attributed to the gradient-drift instabilities associated with the equatorial electrojet (Tsunoda, 2008). At high magnetic latitudes, the vertical motion of gravity waves is very efficient in concentrating the ionization of $E_s$ because the magnetic field lines are nearly vertical in the polar gap (Bautista et al., 1998; MacDougall et al., 2000a, b; MacDougall and Jayachandran, 2005). The $E_s$ layer generally has a vertical scale of 1 km or less, but its horizontal scale can extend up to several hundreds of kilometres. Consequently, intense $E_s$ plasma irregularities and their sharp vertical electron density gradients seriously affect radio communications and navigation systems (Pavelyev et al., 2007). Furthermore, these effects on global positioning system (GPS) radio occultation (RO) signals detected by low-Earth-orbit (LEO) satellites can be exploited for lower-level atmospheric and ionospheric global investigations (Rocken et al., 2000; Hocke and Tsuda, 2001; Schreiner et al., 2007; Yue et al., 2010, 2011).

Observations of $E_s$ layers have been widely investigated from ground-based radars (e.g., Farley, 1985; Whitehead, 1989; Kelly, 1989; Chu and Wang, 1997; Mathews, 1998). In addition to ground-based radars, the scintillations of GPS RO were employed to extensively investigate $E_s$ layers over the past decades (Wu et al., 2005; Arras et al., 2008; Zeng and Sokolovskiy, 2010; Chu et al., 2011). A global map of $E_s$ layers was first presented based on a meridional chain of ionosonde stations (Leighton et al., 1962). In recent years, based on GPS RO measurements, knowledge of the global $E_s$ layer occurrence rate (hereafter called $E_s$OR) has been remarkably advanced. Wu et al. (2005) used phase and signal-to-noise ratio (SNR) variations from ∼6000 GPS/Challenging Minisatellite Payload (CHAMP) occultations to study the global climatology of $E_s$OR. Arras et al. (2008) investigated the global $E_s$OR distribution, with a resolution of $5° \times 5°$, based on CHAMP, GRACE (Gravity Recovery and Climate Experiment), and COSMIC (Constellation Observing System for Meteorology, Ionosphere, and Climate) occultation data. These previous studies of global $E_s$OR maps show a strong seasonal variation, with a summer maximum in the midlatitudes. Chu et al. (2014) employed COSMIC measurements to study the global morphology of $E_s$OR and the results of the theoretical simulations suggested that the $E_s$OR seasonal variation is likely attributed to the convergence of the metallic ion flux caused by vertical wind shear. Shinagawa et al. (2017) calculated the global distribution of the vertical ion convergence and showed that local and seasonal variations in the wind shear distribution could partially account for the geographical and seasonal variation in $E_s$OR.

Many papers have reported the geographical distribution and seasonal variation in global $E_s$ layers retrieved from GPS RO signals, and nearly all of these works focused on the $E_s$OR. The global climatology of the intensity of $E_s$ layers has not been fully studied. The purpose of the present paper is to study the global intensity of $E_s$ layers and compare the results of $E_s$ intensity with previous studies on the $E_s$OR. The occurrence of $E_s$ layers can cause both SNR fluctuations and relative slant total electron content (TEC) peaks (Yue et al., 2015). Sometimes, the SNR has specific U-shape structures in the amplitude of GPS RO signals, as reported by Zeng and Sokolovskiy (2010); Yue et al. (2015). The obvious increase in slant TEC occurring at approximately 92 km implies ionization enhancement in $E_s$. In this study, the scintillation index (S4 index) data measured from SNR fluctuations in the L1 channel of the COSMIC GPS RO profiles at altitudes between 90 and 130 km for the period from December 2006 to January 2014 are employed to study the global climatology of the ionization of $E_s$ layers. Section 2

describes the used data sets and procedures adopted to derive the S4 index. In Section 3, the global long-term behaviours of $E_s$ layers with a high spatial resolution are presented and compared with the previous $E_s$OR results, including the latitude-day, latitude-longitude and altitude-latitude distributions; seasonal variations; and geomagnetic dependence of $E_s$ layers. In Section 4, on the basis of the wind shear theory combined with several global-scale models, namely the Whole Atmosphere

Community Climate Model (WACCM) (Marsh et al., 2013), the Naval Research Laboratory (NRL) Mass Spectrometer and Incoherent Scatter (MSIS)-00 atmospheric model (Picone et al., 2002), and the International Geomagnetic Reference Field (IGRF)-12 geomagnetic field model (Thébault et al., 2015), we calculate the global distribution of the divergence of metallic ion velocity for comparison with the observations of $E_s$ layers from COSMIC satellites. The effect of the magnetic declination angle on the divergence of the metallic ion velocity in the simulation of $E_s$ is investigated for the first time. Section 5 presents

the discussion and conclusions of this paper.

## 2 Data and Procedure of Deriving the S4 Index

The COSMIC global data sets used in this study are the COSMIC-GPS amplitude S4 indices. The GPS radio signals are received by the precise orbit determination antennas of COSMIC for each GPS RO when a GPS sets or rises behind Earth's atmosphere, as seen by the LEO satellite. Once the GPS signal is received at the LEO satellite, the onboard algorithm of the

GPS receiver measures SNR intensity fluctuations from the raw 50 Hz L1 amplitude measurements, which are then recorded in the data stream at a 1 Hz rate at the ground receiver to minimize the data record size (Syndergaard et al., 2006). The raw scintillation measurements from the receiver are therefore the root mean square (RMS) of the SNR intensity fluctuation in one second (i.e., $\sigma_I$), which can be expressed as $\sigma_I = \sqrt{\langle (I - \langle I \rangle)^2 \rangle}$. $I$ represents the square of the L1 SNR, and the bracket $\langle \rangle$ denotes the one second averaged value. The S4 indices are reconstructed by the COSMIC Data Analysis and Archive Center

(CDAAC) ground processing after these $\sigma_I$ data are downloaded (Rocken et al., 2000). During the procedure of deriving the S4 indices, two additional steps are included in the ground processing. The first step is to assume that the SNR intensity fluctuations have Gaussian distributions to calculate an approximate value of $\langle I \rangle$ from $\sigma_I$ and $\langle SNR \rangle$. The second is to apply a low pass filter to the time series of $\langle I \rangle$ to obtain a new average of the intensity $\langle I \rangle_{new}$ at each second to replace the $\langle I \rangle$ in the calculation of the S4 indices. After these steps, a long-term detrended S4 index can be reconstructed by CDAAC ground

processing. Further details on the procedure of deriving the S4 index along with some individual example figures can be found in the report of Ko and Yeh (2010).

In the present study, the COSMIC global data sets specifically denote the maximum value of S4 (S4max). The COSMIC global S4max data include the S4max value and geographic latitude, longitude, altitude and local time on which S4max was detected. The computed detrended S4max index is available from 28 December 2006 onwards on the CDAAC website

(http://cdaac-www.cosmic.ucar.edu/cdaac). The long-term global climatology of the $E_s$ intensity is investigated based on the global S4max data from December 2006 to January 2014. Figure 1 shows the altitude distribution of the COSMIC S4max profiles. A considerable number of profiles are distributed at altitudes between 40 and 130 km, with a peak number at approximately 100 km. Information on $E_s$ layers can be extracted from amplitude fluctuations in the SNR profiles (Wu et al., 2005).

Please note that as a result of the integrated influence either in the SNR or the slant TEC along the LEO-GPS ray, the effect of $E_s$ layers at high altitudes could map down to the lower tangent point altitudes, which may virtually induce multiple peaks in one RO event (Zeng and Sokolovskiy, 2010; Yue et al., 2015). In Figure 1, the occurrence of sporadic E can be seen down to 40 km as a result of the integral problem of RO measurements. In fact, $E_s$ layers could not be formed by the wind shear theory below 90 km because of the high ion-neutral collision frequencies. The $E_s$ layer over lower altitudes (between 40 and 90 km) should be some artefact resulting from the mapping effect integrated along the LEO-GPS ray. Therefore, the S4max values that appear in altitudes ranging between 90 and 130 km are used to study the $E_s$ layers in the lower ionosphere region. Figure 2 shows the entire distribution of the daily average Es intensity from 2006 to 2014 retrieved from COSMIC within $\pm\,2.5°$ latitude and longitude of one ionosonde station in Beijing ($40.3°N, 116.2°E$), which agrees with the ionosonde measurements ($f_oEs$). The global morphology of the $E_s$ intensity is presented, and its altitude and seasonal dependences are given at a high spatial resolution because of the large COSMIC RO data sets which have high vertical resolutions.

## 3   Observations

Figure 3 shows the long-term time series of the $E_s$ values, with a resolution of 5° latitude $\times$ 5 days. As shown, the $E_s$ layer is mainly a sporadic-layered phenomenon in the summer hemisphere, as is known from former $E_s$OR studies (Leighton et al., 1962; Wu, 2006; Arras et al., 2008; Chu et al., 2014). In Figure 3, it is clear that the intensity of $E_s$ is enhanced in the northern (southern) summer hemisphere from May to September (from November to March), with a maximum in June (December) (i.e., one month ahead of the $E_s$OR maximum (Chu et al., 2014)). In addition, the seasonal $E_s$ layer also has interannual variability. Compared with the intense Es activity in the summers of 2010 and 2011, the intensity of $E_s$ is lower in the northern summers in 2012 and 2013, respectively. This result may be caused by anomalies in the wind field in the upper atmosphere and a corresponding reduction in the vertical wind shear associated with $E_s$ formation.

The map in Figure 4 shows the global geographical distributions of the $E_s$ average intensity, with a significantly improved spatial resolution of $1° \times 1°$. The red and green solid curves represent the northern and southern geomagnetic latitude contours of 60°, 70°, and 80°, respectively. The geomagnetic equator is also plotted with a yellow curve. The $E_s$ layers are predominantly distributed with S4max values exceeding 0.7 at midlatitudes. Because of the increased spatial resolution, the regional features and longitudinal variations become visible. The intensity of $E_s$ is much weaker at the lower latitudes in both hemispheres, especially in the noticeable gap near the magnetic equator. When the magnetic field is horizontal at the geomagnetic equator, under the zonal wind shear action, ions move vertically by Lorentz forcing. However, they fail to converge into a layer because they are withheld by magnetized electrons. The plasma remains locally neutral. For a meridional wind shear process, ions move along the magnetic field lines with no Lorentz forces acted upon (Haldoupis, 2012). Therefore, a noticeable gap near the magnetic equator could be expected, which is explained by the vanishing vertical component of the geomagnetic field lines, keeping the ionized particles from effectively vertically converging. This gap could also be found in the distribution of $E_s$OR although it is not as evident in Arras et al. (2008).

Furthermore, the $E_s$ longitudinal variations in the geomagnetic field are also clearly shown. The decrease in $E_s$ intensity can be seen clearly in the Southern Atlantic Anomaly (SAA) zone and the northern American region with geomagnetic latitude. The region of large $E_s$ intensity exists in the North Africa and North Atlantic regions, Southeast Asian region, South Africa and South Pacific regions. A difference between the $E_s$ intensity and $E_s$OR distributions is at high latitudes: that is, the occurrence rates of $E_s$ are generally low (Arras et al., 2008), but the intensity of $E_s$ is relatively high. This pattern is more evident over the magnetic poles, which is likely the result of vertical motions of gravity waves in concentrating the ionization of $E_s$ layers (MacDougall et al., 2000a, b). The lower panels depict the northern and southern polar views of the distributions of $E_s$ intensity, and these views make the features clearer.

The maps in Figure 5 show the geographical distribution of $E_s$ intensity for four different seasons in a $1° \times 1°$ grid. The distribution of $E_s$ layers shows a significant seasonal dependence. The intensity of the $E_s$ layers in the midlatitudes of the summer hemisphere is 2–3 times larger than that in the winter hemisphere. During equinox seasons, the intensity of the $E_s$ layers moderately covers the entire globe, and a distinct boundary dividing the midlatitudes and high latitudes is visible along the $60°–80°$ geomagnetic latitude band. From the polar views during each season, it can be seen that the $E_s$ layers remain at a relatively high level in the North and South Poles. This characteristic could be attributed to high energy radiation, particle precipitation and polar gap gravity waves.

Figure 6 shows the altitude-latitude distribution of the $E_s$ intensity with a resolution of 1 km altitude $\times$ $1°$ latitude. The intensity of $E_s$ is distributed at altitudes between 95 and 125 km. The densest patches of $E_s$ exist at altitudes exceeding 110 km, which is different from the $E_s$OR altitude-latitude distribution that dominates at 95–110 km, with a peak at approximately 105 km in the midlatitudes of $25°–45°$(Arras et al., 2008). The $E_s$ intensity has a broader latitudinal extent of $10°S–75°S$ in the Southern Hemisphere, compared with $10°N–60°N$ in the Northern Hemisphere.

Figure 7 presents the seasonal variation in the altitude-latitude distributions of the $E_s$ intensity for the same temporal period and spatial resolution as those in Figure 6. The $E_s$ intensity for the summer and winter solstices clearly has a significantly broader latitudinal extent towards the high latitude region. In addition, the overall intensities of the $E_s$ layers increase, spanning a larger vertical extent during the solstices. In general, the $E_s$ intensity exceeding 0.65 values is distributed at altitudes of 100–125 km in the southern summer and at altitudes of 90–130 km in the northern summer. During equinox seasons, the $E_s$ intensity is moderate and its altitude-latitude distribution is relatively symmetric.

## 4 Wind Shear Theory Explanation for $E_s$ Seasonal Variation

The global climatology of the intensity of $E_s$ layers is investigated from the COSMIC occultation data employing the GPS RO technique. One of the pronounced variabilities in $E_s$ layers is the seasonal variation, with a maximum appearance in the summer hemisphere. Although the mechanism for $E_s$ layer formation is widely accepted, these dense and thin layers of metallic ion plasma are formed by the vertical ion convergence of neutral wind shear. The overall morphology of $E_s$ layers cannot be explained by the wind shear theory. One of the unsolved issues in the ionosphere is that the overall morphology, including the seasonal dependence of Es layers, does not have a comprehensive explanation (Whitehead, 1989; Haldoupis et al., 2007).

Seasonal dependence is found not only in the $E_s$ intensity but also in previous studies of $E_s$OR variations (Wu et al., 2005; Arras et al., 2008; Chu et al., 2014). Chu et al. (2014) simulated the global distribution of the convergence of metallic ion flux caused by vertical wind shear, suggesting that the maximum $E_s$ in summer and minimum of $E_s$ in winter are likely caused by the vertical wind shear effect.

From the wind shear theory (e.g., Nygren et al., 1984; Mathews, 1998; Kirkwood and Nilsson, 2000), the vertical ion velocity $w_i$ induced by the neutral wind is described by equation (1):

$$w_i = \frac{r_i cosI}{1+r_i^2} \times U - \frac{sinIcosI}{1+r_i^2} \times V + \frac{r_i^2 + sin^2I}{1+r_i^2} \times W \qquad (1)$$

where $I$ represents the magnetic inclination angle that is defined as positive (i.e., downward direction) in the Northern Hemisphere, $r_i$ represents the ratio of the ion-neutral collision frequency ($\nu_i$) to the ion gyrofrequency ($\omega_i$), and the neutral

wind velocity $V_n = (U, V, W)$ components are in the zonal (positive for eastward), meridional (positive for northward), and vertical (positive for upward) directions. Therefore, the favourable wind field for $E_s$ layer formation is where there is a negative $\frac{dw_i}{dz}$ relationship, indicating an ion-convergence region.

    Version 4 of the WACCM (WACCM4) is a global climate model with interactive chemistry, developed at the National Center for Atmospheric Research (NCAR) (Marsh et al., 2013). A specified dynamics run of WACCM4 (SD-WACCM4) was

constrained by the Modern-Era Retrospective Analysis for Research and Applications (MERRA). SD-WACCM4 is used to simulate the global distribution of the divergence of ion velocity from the period of 2006 to 2014, which is consistent with the period of $E_s$ observations from the COSMIC occultation data. To compare with previous studies, the neutral wind is provided by the output from WACCM, and the ion-neutral frequency is calculated by the atmospheric composition from the MSIS-00 atmospheric model in accordance with Chu et al. (2014). The global distributions of the geomagnetic field and magnetic

inclination angle at 100 km are estimated from the IGRF-12 model. The calculation of ion velocity is binned and averaged in a $1°$ latitude $\times$ $1°$ longitude grid at WACCM altitude levels from 0–130 km.

    Figure 8 presents simulation results of the global distributions of the monthly mean divergence in vertical ion velocity in the altitude range between 97 and 114 km in January and July. The negative (positive) $\frac{dw_i}{dz}$ ratio represents the convergence (divergence) of ions in units of $ms^{-1}km^{-1}$. Ihe results show a good correlation between the simulated distributions of the

monthly mean divergence of vertical ion velocity in Figure 8 and the geographical distribution of $E_s$ intensity measured from the COSMIC GPS RO profiles in Figure 5. Chu et al. (2014) simulated the global distributions of the mean divergence of the $Fe^+$ concentration flux at altitudes of 94–115 km in all four seasons. The study also showed a similar simulation result of the distributions of divergence in the $Fe^+$ concentration flux, which is well correlated with the COSMIC-measured $E_s$OR distribution. The simulation of the divergence of vertical ion velocity supports the wind shear theory for $E_s$ formation and

indicates that the seasonal dependence of $E_s$ layers is likely attributed to the convergence of vertical ion velocity driven by neutral wind.

    Furthermore, we also notice that the $E_s$ intensity is distributed at relatively higher altitudes of 95–125 km compared with the $E_s$OR at 90–120 km. The densest patches of $E_s$ exist above 115 km, and the $E_s$ layer has a broader vertical extent in summer,

as shown in Figure 7. In the simulation, the distributions of the monthly mean divergence of the vertical ion velocity at an altitude range of 114–128 km in January and July are shown in Figure 9. In contrast to the distributions of the divergence of vertical ion velocity between 97 and 114 km in Figure 8, Figure 9 shows an ion-divergence region at altitudes of 114–128 km in summer over midlatitudes as a result of different zonal and meridional winds. These results suggest that the wind shear theory

alone has difficulty explaining the $E_s$ seasonal dependence at higher altitudes (114–128 km), although the wind shear theory is considered the primary theory to explain the physical production of $E_s$ layers (Whitehead, 1989; Haldoupis et al., 2007).

In previous studies on the wind shear theory for $E_s$ layer formation, the magnetic declination angle effect is neglected in the calculation of the vertical ion velocity $w_i$ induced by the neutral wind. The steady-state ion momentum equation is:

$$m_i \frac{\mathrm{d}\boldsymbol{v}_i}{\mathrm{d}t} = 0 = e(\boldsymbol{E} + \boldsymbol{v}_i \times \boldsymbol{B}) - m_i \nu_{in}(\boldsymbol{v}_i - \boldsymbol{V}_n) \tag{2}$$

On the basis of the steady-state ion momentum equation, the equation (1) for the vertical ion velocity $w_i$ is extended to take the magnetic declination angle $D$ into consideration as follows:

$$w_i = \frac{r_i cosDcosI - sinDsinIcosI}{1 + r_i^2} \times U - \frac{r_i sinDcosI + cosDsinIcosI}{1 + r_i^2} \times V + \frac{r_i^2 + sin^2 I}{1 + r_i^2} \times W \tag{3}$$

The magnetic declination angle currently ranges from -30° (west) to 26° (east); therefore its influence on the vertical ion velocity $w_i$ is expected. The effect of the magnetic declination angle on the divergence in ion velocity in the simulation of $E_s$

is investigated. Figure 10 presents the global distributions of the monthly mean divergence in vertical ion velocity at altitudes ranging from 97–114 km with the consideration of the magnetic declination angle. The figure shows a seasonal dependence, with ion-convergence regions in summer and ion-divergence regions in winter. However, the morphology of the divergence of the vertical ion velocity is different from that without the magnetic declination angle considere in Figure 8. In January, strong ion convergence appears in the SAA region. In July, Asia, Europe, and the North Pacific tend to be regions of ion convergence.

The agreement with the observations becomes worse, which could imply that the cause of global $E_s$ layers remains a mystery because it cannot be fully accounted for by the wind shear effect (Whitehead, 1989; Haldoupis et al., 2007). The formation of mid-latitude $E_s$ layers could be partially explained by the wind shear theory. The investigation of the causes of seasonal variability in $E_s$ should lead to more detailed studies to fully understand and properly quantify the properties of $E_s$ layers.

## 5   Discussion

The seasonal and geographical dependences of $E_s$OR have been widely studied by ionospheric observations since 1960s (Leighton et al., 1962; Smith, 1978; Wu et al., 2005; Arras et al., 2008; Zeng and Sokolovskiy, 2010), but, thus far, the overall morphology of $E_s$ is still not well explained. The seasonal dependence of $E_s$ layers remains an ongoing mystery, as it is unexpected in the classical wind shear theory reported in the review article of Whitehead (1989). Recently, Chu et al. (2014) simulated the distribution of the convergence of the $Fe^+$ concentration flux and indicated that the vertical ion convergence

caused by neutral wind could be responsible for the seasonal dependence of $E_s$.

In our investigations, the global climatology of the intensity of $E_s$ layers is found to also have a seasonal dependence, with a pronounced maximum over midlatitudes in the summer hemisphere, as shown in Figure 5. The $E_s$ intensity has similar seasonal and spatial distributions as the $E_s$OR, but the $E_s$ layer has a relatively large intensity and a small $E_s$OR value at the North and South Poles. The wind shear mechanism does not work efficiently at either auroral zones or the magnetic equator (Haldoupis,

2012); therefore, the strong $E_s$ layers in the Earth's polar regions could be initially caused by gravity waves (Bautista et al., 1998; MacDougall et al., 2000a, b). In the simulations, the gravity waves with horizontal wavelengths smaller than ∼200 km are not explicitly resolved in WACCM (Liu et al., 2014). In particular, the vertical motion of gravity waves dominates the formation of the $E_s$ layer in the polar cap, where the near-vertical magnetic field significantly reduces the effectives of wind-shear in converging ions into layers. Polar cap gravity waves have been studied by Johnson et al. (1995); MacDougall et al.

(1997). These layers are maintained in an ionized state by charge exchange of neutral metal atoms with $NO^+$ and $O_2^+$ ions by photoionization. These studies found that the vertical motion of gravity waves is very efficient in concentrating polar cap $E_s$ layers. The short-lived polar cap $E_s$ layers in winter appear to be associated with gravity waves. The polar cap $E_s$ layers in summer are long-lasting thin layers. These initial concentrations of metallic ions persist and change into long-lived $E_s$ because of ambient metallic ions produced by photo-ionization in the sunlit E region (MacDougall et al., 2000a). The $E_s$ layers at the

cusp latitude are relatively different from those at the polar cap. The cusp $E_s$ could be associated with the convergence of ionization by the electric fields (MacDougall and Jayachandran, 2005).

On the other hand, simulating the global distributions of the monthly mean divergence of vertical ion velocity in an altitude range between 97 and 114 km shows an ion-convergence region in the summer midlatitudes, which is similar to the simulation results of Chu et al. (2014). This result suggests that the seasonal dependence of $E_s$ is likely attributed to the vertical conver-

gence of ions driven by neutral wind. However, some disagreements between the distributions of the calculated divergence of vertical ion velocity and observed $E_s$ intensity are found. For example, there are ion-divergence regions in the midlatitudes in winter in Figure 8, but the dissipation of $E_s$ is observed in the $60°$–$80°$ geomagnetic latitude band. The densest $E_s$ layer appears above 115 km, which is higher than the $E_s$OR. Another discrepancy is that the simulated divergence of vertical ion velocity in an altitude range between 114 and 128 km has a positive $\frac{dw_i}{dz}$ ratio in the summer hemisphere, which indicates an

ion-divergence region of ions in contrast to the observed summer maximum of $E_s$ intensity in the summer midlatitudes.

The effect of the magnetic declination angle on the divergence of metallic ion velocity is investigated in the simulation of $E_s$ for the first time in Figure 10. Though the figure shows marked seasonal dependence, with a strong summer ion-convergence region, the morphology of the divergence of vertical ion velocity is different from the distribution of the observed $E_s$ intensity in Figure 5. Thus, the vertical ion convergence by itself is far from sufficient for explaining the strong $E_s$ summer maximum.

Other physical processes should also be considered in the geographical distribution and spatial variations in $E_s$ layers, as they play important roles in determining the global morphology of $E_s$ such as the magnetic field, ionospheric electric field, the chemical processes of metallic ions, large geomagnetic storms, and meteorological processes in the lower atmosphere (e.g., Bautista et al., 1998; Mathews, 1998; Carter and Forbes, 1999; MacDougall et al., 2000a, b; Davis and Johnson, 2005; MacDougall and Jayachandran, 2005; Johnson and Davis, 2006; Haldoupis, 2012; Yue et al., 2012; Feng et al., 2013; Yu et al.,

35   2015).

Haldoupis et al. (2007) proposed that the seasonal dependence of $E_s$ could be explained by the seasonal variation in the meteor influx into the upper atmosphere. However, it has been largely accepted that sporadic meteoroids provide a much greater meteor mass on average than meteor showers (Ceplecha et al., 1998; Baggaley, 2002; Janches et al., 2002; Williams and Murad, 2002). The meteoric mass influx caused by sporadic meteoroids reaches a maximum in autumn rather than summer (Janches et al., 2006). The global input of meteoric material is well established to enhance the mesospheric metal layers and $E_s$ layers (Plane, 2004; Plane et al., 2015). Carrillo-Sánchez et al. (2015) estimated the global interplanetary dust particles (IDPs) $43\pm14$ tons per day, from three observations to constrain the relative contributions of each dust source with lidar observations of the vertical Na and Fe fluxes. However, the daily amount is still not well defined. Estimates of the IDP range from 5–270 tonnes per day, depending on the different methods used to make the estimate (Plane, 2012). These effects of meteoric ablation are significantly influenced by the magnitude of the IDP input by two orders of magnitude uncertainty. On the other hand, this fact also highlights the importance of fundamental understanding of the global climatology of $E_s$ layers.

## 6    Conclusions

In this study, we investigate the long-term climatology of the intensity of $E_s$ layers on the basis of S4max data retrieved from COSMIC GPS RO measurements. The resulting global $E_s$ maps with a high spatial resolution present the geographical distributions and strong seasonal dependence of $E_s$ intensity, which agrees with former studies of global $E_s$OR maps (Wu, 2006; Arras et al., 2008; Chu et al., 2014). The high $E_s$ intensity in summer exists at altitudes of 115–125 km at 10°–60° latitude in the Northern Hemisphere, and at altitudes of 115–120 km in the 10°–75° latitudes in the Southern Hemisphere.

Furthermore, the simulation results of the global distributions of the monthly mean divergence of vertical ion velocity could partially explain the seasonal dependence of $E_s$ intensity. We show that the elemental mechanism responsible for $E_s$ layers based on the wind shear theory could explain the seasonal dependence of Es intensity (97–114 km), but it is hard to explain the Es seasonal dependence at higher altitudes (114–128 km). To further investigate the magnetic field effects on the wind shear processes of $E_s$ formation, the effect of the magnetic declination angle on the divergence of metallic ion velocity in the simulation of $E_s$ is investigated, and we discuss some disagreements between the distributions of the calculated divergence of vertical ion velocity and the observed $E_s$ intensity. Although the wind shear theory for the $E_s$ formation was conceived and formulated in 1960s (Whitehead, 1961), its importance for understanding the formation of $E_s$ must have escaped attention. This study implies that, in addition to the vertical wind shear effects, other processes, such as the vertical motion of gravity waves, magnetic field effects, meteoric mass influx into Earth's atmosphere and the chemical processes of metallic ions, should also be considered, which could play a dominant role in the geographical and seasonal variations in $E_s$ layers. To accurately understand and properly quantify the properties of $E_s$ layers at a global scale that are also associated with the distribution of global metallic ions, we need to combine more ground-based ionosonde data with satellite observations and extensively study the geographical and seasonal variations in $E_s$ layers.

*Author contributions.* BY and XX designed the study and wrote the manuscript. XY provided the COSMIC radio occultation data and contributed significantly to the comments on an early version in the manuscript. CY and CY discussed the results of the wind shear theory simulation. BN and LH provided the manually scaled ionospheric observation at Beijing. XD contributed to the discussion of the results and the preparation of the manuscript. All authors discussed the results and commented on the manuscript at all stage.

5   *Competing interests.* The authors declare that they have no conflict of interest.

*Acknowledgements.* We acknowledge the COSMIC (Constellation Observing System for Meteorology, Ionosphere, and Climate) radio occultation data, the ionosonde data from the Chinese Meridian Project, the Solar-Terrestrial Environment Research Network (STERN), Data Center for Geophysics, Data Sharing Infrastructure of Earth System Science, National Science & Technology Infrastructure of China as well as the the Whole Atmosphere Community Climate Model (WACCM), NRL Mass Spectrometer and Incoherent Scatter (MSIS)-00 atmospheric model, and International Geomagnetic Reference Field (IGRF)-12 geomagnetic field model data used in this paper. This work is supported by the National Natural Science Foundation of China (41774158, 41474129, 41421063,41804147), the open research project of CAS Large Research Infrastructures, the Youth Innovation Promotion Association of the Chinese Academy of Sciences (2011324) and the Fundamental Research Fund for the Central Universities. BY is also supported by a Newton International Fellowship from the Royal Society.

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

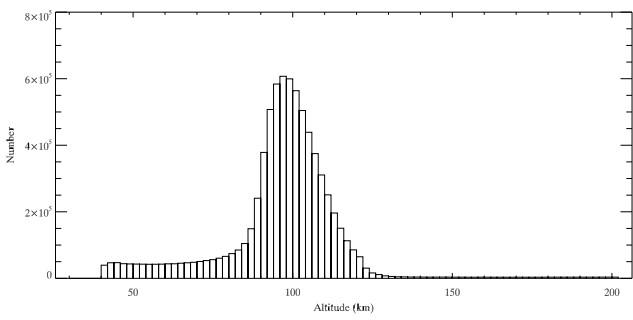

**Figure 1.** The altitude distribution of COSMIC S4max profiles from December 2006 to January 2014.

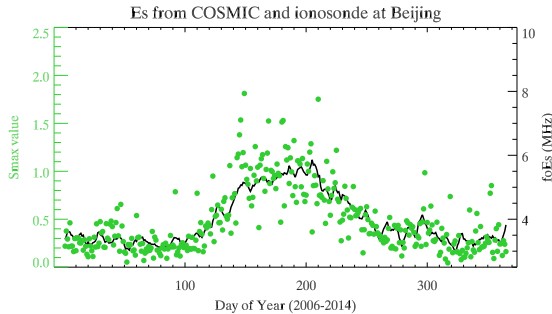

**Figure 2.** The entire distribution of the daily average Es intensity retrieved from COSMIC within $\pm\ 2.5°$ latitude and longitude of one ionosonde station and ionosonde data ($f_oEs$) in Beijing from 2006 to 2014.

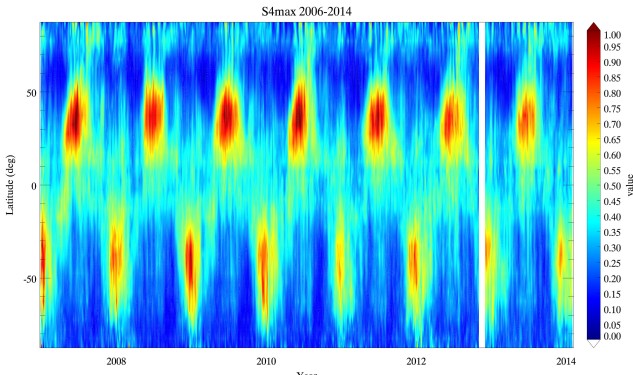

**Figure 3.** Time series of the $E_s$ intensity with a resolution of 5° latitude × 5 days for the period from December 2006 to January 2014.

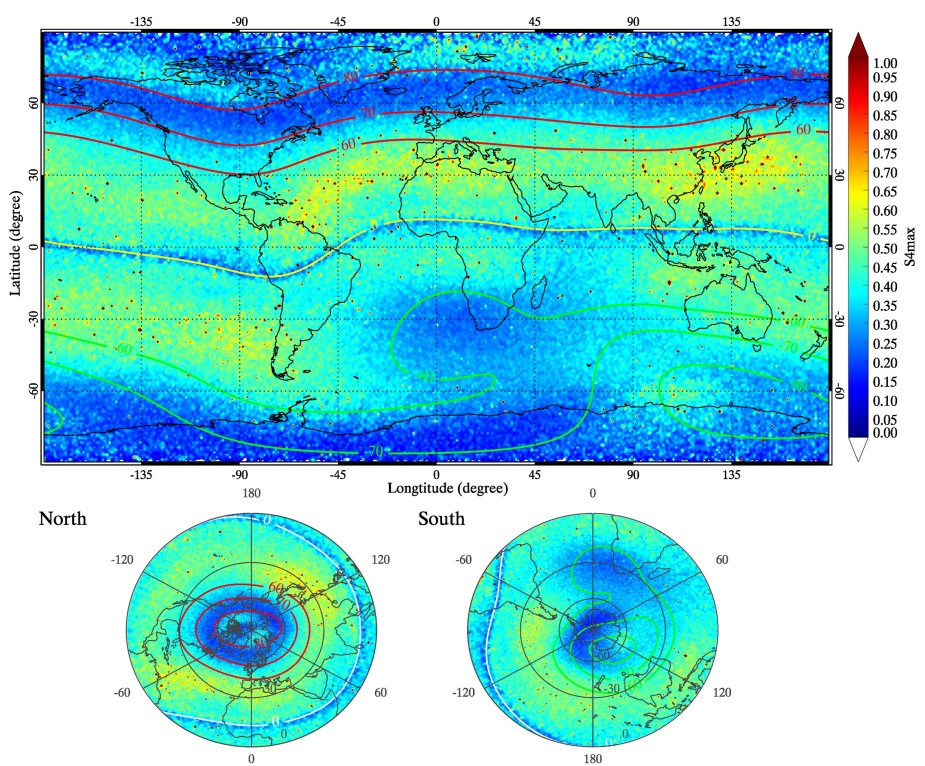

**Figure 4.** Global geographical distributions of the $E_s$ average intensity from 2006 to 2014, with a spatial resolution of a $1° \times 1°$ grid. The red and green curves signify the geomagnetic latitude contours of $60°$, $70°$, and $80°$, and the yellow curve represents the geomagnetic equator.

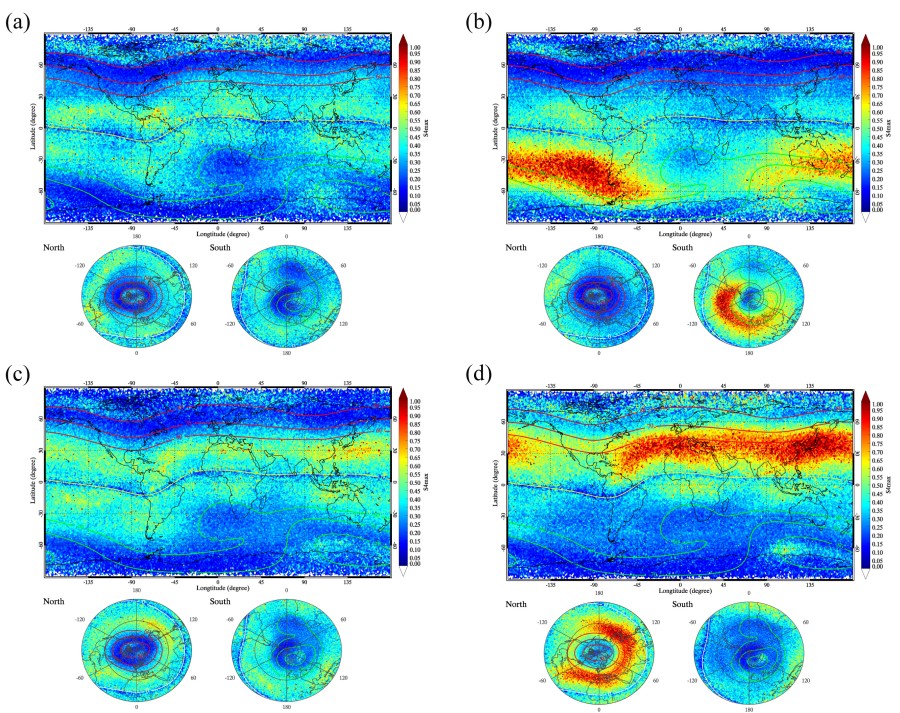

**Figure 5.** Seasonal variations in $E_s$ intensity from 2006 to 2014, with a spatial resolution of a $1° \times 1°$ grid. Plots for (a) autumn (September, October, November), (b) winter (December, January, February), (c) spring (March, April, May), and (d) summer (June, July, August).

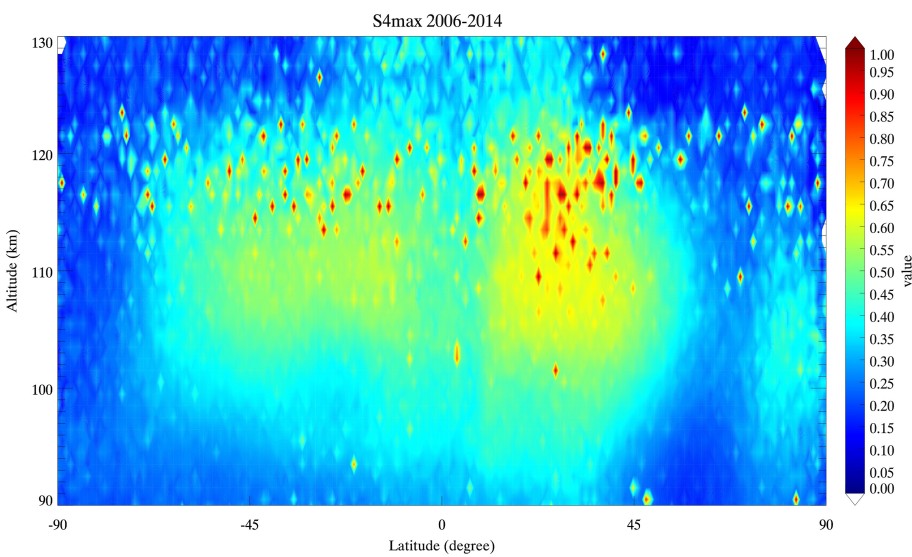

**Figure 6.** Altitude-latitude distribution of the $E_s$ intensity from 2006 to 2014, with a resolution of 1 km altitude $\times$ 1° latitude.

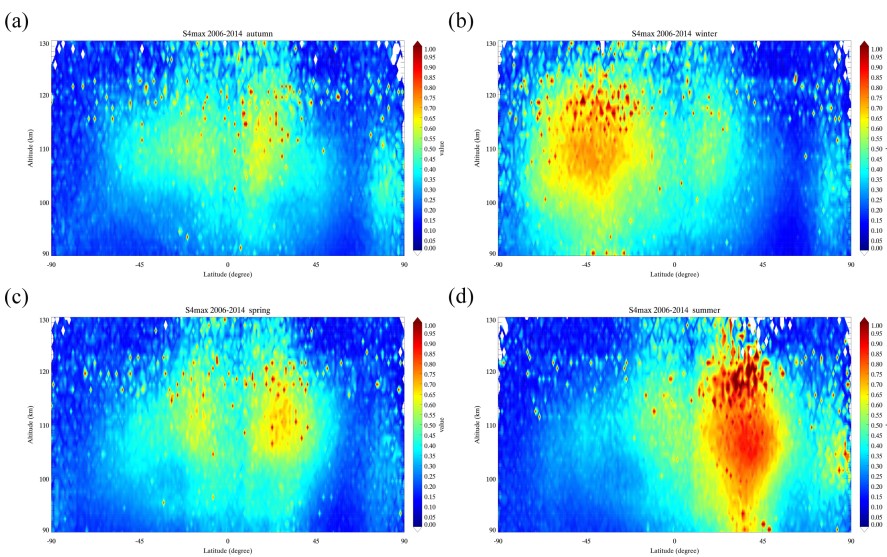

**Figure 7.** Seasonal variations in altitude-latitude distributions of $E_s$ intensity from 2006 to 2014 for four different seasons: (a) autumn, (b) winter, (c) spring, and (d) summer.

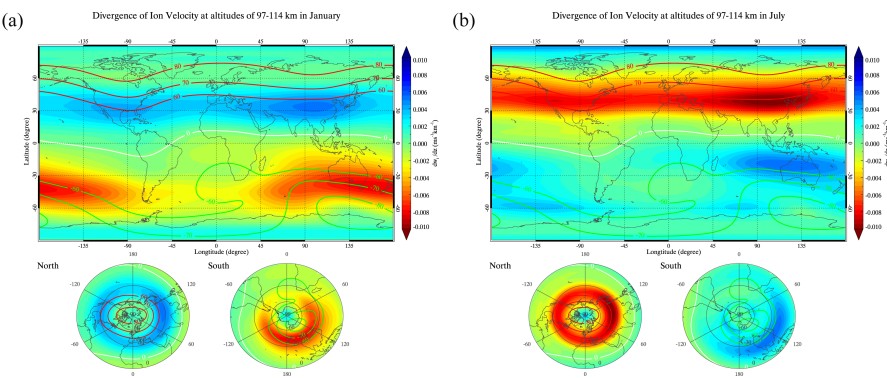

**Figure 8.** Simulation results of the global distributions of the monthly mean divergence of vertical ion velocity from 2006 to 2014 (units of $ms^{-1}km^{-1}$) at altitudes ranging between 97 and 114 km in January (a) and July (b). The red and green curves signify $60°$, $70°$, and $80°$ geomagnetic latitude contours, and the yellow curve represents the geomagnetic equator.

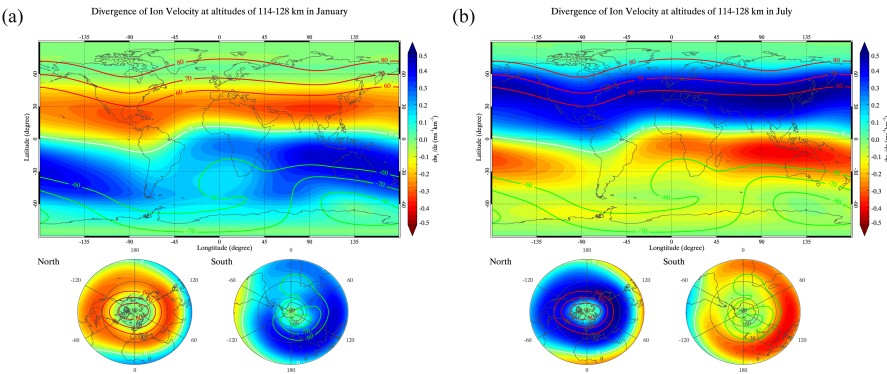

**Figure 9.** Same as Figure 8 but for the altitude range between 114 and 128 km in January (a) and July (b).

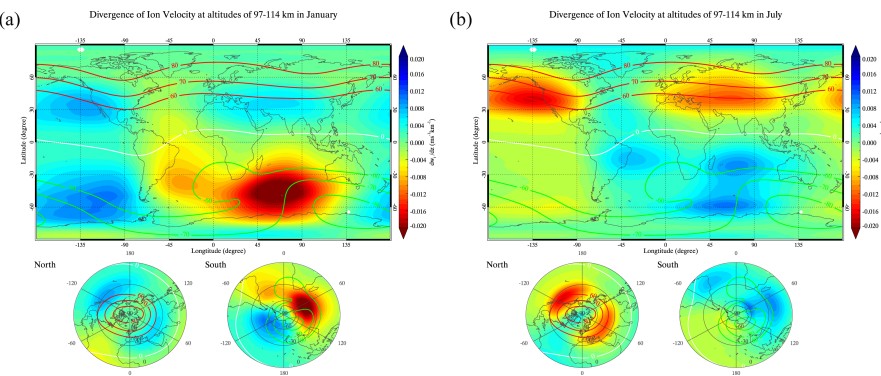

**Figure 10.** Same as Figure 8 but with the consideration of the effect of the magnetic declination angle on the vertical ion velocity in January (a) and July (b).