# Peer review of "The global climatology of the intensity of the ionospheric sporadic *E* layer"

_Atmospheric Chemistry and Physics, 2018_

## Referee Comment (RC1) · Anonymous Referee #1 · 31 Oct 2018

This paper uses the same methodology developed in a previous study (Chu et al., 2016) for deriving sporadic E properties from COSMIC GPS data. This study goes further by examining both the occurrence frequency and strength, as a function of season, latitude, longitude and height. A number of interesting observations are made. The authors then show that some of these observations can be explained by using winds from a global chemistry-climate model (WACCM) to calculate wind shear and hence ion convergence.

One curious omission is discussion of gravity waves, which are not explicitly resolved in WACCM if their horizontal wavelengths are smaller that $\sim$200 km. Many of these waves penetrate into the lower thermosphere, and appear to be a major cause of sporadic E layers in the polar cap, where the near-vertical magnetic field reduces significantly the

effectives of wind-shear in converging ions into layers. The authors appear to be using winds averaged over an unspecified period to determine wind shear – yet there is no discussion of how valid this is, since sporadic E often have short lifetimes of only hours.

Another point is that the height distribution of sporadic E (Figure 1) shows a relatively large proportion of layers appearing between 40 and 90 km. This is dismissed here as an artefact, based on the fact that sporadic E should not form below 90 km because the ion-neutral collision frequency is too high (page 3). The explanation for the artefact is unclear – reference is made to an "RO" event, though this term is not defined in the paper – but appears to be caused by slant viewing geometries. If that is the case, how reliable is the entire distribution of layers, including those above 90 km? The authors ought to show that the distribution they are using agrees with ionosonde measurements at a particular location.

The paper ends on a vague and rather disappointing note: "It indicates that, in addition to the vertical windshear effects, other processes such as magnetic field effects, meteoric mass influx into the earth's atmosphere and chemical processes of metallic ions are also likely to play an dominant role in the geographical and seasonal variations of Es layers." There is no attempt to explain how these "chemical processes", the meteoric mass influx, or unspecified "magnetic field effects" could explain the observations which do not accord with the wind shear theory.

Specific points which need to be addressed: p. 4, line 10: why do you state "may be caused"? You have applied several different models, including WACCM. What do they tell you about the anomalies? The specified dynamics version of WACCM would be quite informative. You also give no details about the version of WACCM output that you use, etc.

p. 4, line 17: surely the wind shear mechanism should be most effective at the geomagnetic equator when the magnetic field is horizontal? Why do you state that a vertical component is required for ion convergence?

p. 4, line 32: the sentence "The ionization of Es layers is persistently magnetic fields trapped in the polar regions" make no sense.

p. 5, line 18: the sentence "One of the unsolved issues in the ionosphere is that the well pronounced seasonal dependence of mid-latitude Es layers does not have a comprehensive explanation, which is inexplicable from the windshear theory" contradicts your own conclusion that wind shear does explain many of the mid-latitude features!

p. 5, line 24: you suddenly mention Fe+ ions here. Why only Fe+, where do they come from? What is the evidence?

p. 6, line 1: why do you take the wind from WACCM, and the atmospheric composition from MSIS? This is inconsistent. This calculation should be performed using composition and winds from the same model.

p. 6, line 19: the windshear theory does not explain formation of Es layers at high geomagnetic latitudes, because the magnetic field is nearly vertical. There is evidence that within the polar cap gravity waves play a dominant role in Es formation (see, e.g. the papers by John MacDougall from Western Ontario). This is not discussed anywhere.

p. 6, line 25: did you derive equation (3) here? If not, it should be referenced.

p. 6, line 32: why are the regions of ion convergence different when the magnetic declination angle is included. You seem to imply that agreement with the observations is worse. What does that imply? Is equation (3) incorrect?

p. 7, line 11: why would energetic particle precipitation and cosmic rays generate Es in the polar region? Most cosmic ray ionization occurs around the tropopause, and EPP would not create a thin layer of ions.

p. 8, line 2: I thought this range of IDP input had been considerably reduced in the Carrillo-Sanchez et al., GRL 2016 paper (which followed the one you cite). What difference would it make anyway to sporadic E formation?

Corrections: Need to indicate in the figure captions for Figure 3, 4, 5, 6, 7, 8 and 9 whether this is data averaged over the whole period or for a single year.

References: many do not have the complete initials of the authors

p. 1, line 3: "show a high . . . distribution and . . ."

p.1, line 6: " . . .bands. Simulations results show that . . ."

p. 1, line 12: ". . .layers are thin-layered . . ."

p. 1, line 19: "equatorial region"

p. 2, line 1: ". . .irregularities and their sharp . . .."

p. 2, line 33: ". . . models, namely the Whole . . ."

p. 3, line 2: ". . . time. Section 5. . ."

p. 3, line 6: ". . .behind earth's . . ."

p. 3, line 7: ". . .signal is received at . . ."

p. 3, line 16: ". . .time series . . ."

p. 6, line 9: ". . . seasons. They also showed . . ."

p. 7, line 31: ". . .provides a much greater . . ."

There are many other grammatical errors in the paper which need to be corrected.
* * *

---

## Referee Comment (RC2) · Anonymous Referee #3 · 8 Jan 2019

This paper reports some novel data and modelling on the occurrence and intensity of global Sporadic E layers and provides some interesting perspectives on the formation mechanisms of these layers. The data are derived from measurements of the S4 index from radio occultation measurements made by the COSMIC satellite constellation and the global distribution of Sporadic E which they reveal is similar to those derived from previous studies, with a strong occurrence peak in the mid-latitudes of the summer hemisphere. The authors comment on some interesting distinctions between occurrence and intensity of sporadic E layers; for example they notes that high-latitude layers, while being lower occurrence, tend to be quite intense when they do arise.

These are interesting data sets, but are somewhat spoiled by their relatively poor presentation. For example it would be nice to have seen graphs of global Sporadic-E

occurrence and global Sporadic-E intensity in the same format, whereas what we actually see is a map of global occurrence (presumably averaged across seasons) in Figure 3 and then separate maps of intensity for each season in Figure 4.

The most interesting part of the study is the attempt to explain the occurrence and intensity of the Sporadic E layers in terms of modelled neutral wind convergence, using wind fields from the WACCM model. This provides qualitative agreement with the observations, if it is assumed that the layers are due to neutral wind convergences in the lower E-region (97-114 km) but strongly suggests that wind convergences at higher E-region altitudes cannot explain the observations. A nice feature of the wind field modelling is that the magnetic declination has been properly included in the calculations and it is demonstrated that allowing for this factor changes the expected distribution of the modelled wind convergences significantly.

Unfortunately, however, the paper is not able to make any firm conclusions, because the correspondence between the modelled wind convergences and the occurrences and intensities of the Sporadic E layers remains only qualitative at best. This almost certainly illustrates the deficiencies of the modelling assumptions. The wind fields, for example, are obviously idealised and must have significantly greater variability than the modelling suggests, an idea reinforced by the inter-annual changes in the occurrence data shown in Figure 2. In addition the authors comment on various other factors such as the variability of the meteor flux, the effects of geomagnetic storms and the effect of meteorological processes, any and all of which could result in differences between the modelling and the observations, but which would be hard to account for without much more complicated modelling. As a result, the interesting features which are observed are not very well explained.

The language of the paper could be improved. It is occasionally imprecise, so that the meaning can be hard to decrypt. There are also some mistakes in spelling and grammar. These are not really what weaken the paper, however. The fundamental problem is that the processes which produce the Sporadic E layers are likely to have

such a complex variability that no simple model can do a good job of characterising them, and this is what the study ultimately shows.

---

## Author Comment (AC1) · 10 Feb 2019

We would like to thank the reviewers for their valuable comments and suggestions. We have studied all comments carefully and these comments have helped us to significantly improve our manuscript. Following the reviewers' comments, we revised the manuscript. Our responses to the reviewers' comments and corresponding changes with page and line numbers in the revised manuscript are both detailed below in blue text. We mark the major changes in the track-change manuscript.

Reviewer #1 comments (RC1): This paper uses the same methodology developed in a previous study (Chu et al., 2016) for deriving sporadic E properties from COSMIC GPS data. This study goes further by examining both the occurrence frequency and

strength, as a function of season, latitude, longitude and height. A number of interesting observations are made. The authors then show that some of these observations can be explained by using winds from a global chemistry-climate model (WACCM) to calculate wind shear and hence ion convergence.

Response: Thank you for your positive comments. Many studies have reported the geographical distribution and seasonal variation in global Es layers retrieved from GPS RO signals, and nearly all of these works focused on the EsOR. The global climatology of the intensity of Es layers has not been fully studied. Our study is to investigate the global intensity of Es layers and compare the results of Es intensity with previous studies on the EsOR.

One curious omission is discussion of gravity waves, which are not explicitly resolved in WACCM if their horizontal wavelengths are smaller that ∼200 km. Many of these waves penetrate into the lower thermosphere, and appear to be a major cause of sporadic E layers in the polar cap, where the near-vertical magnetic field reduces significantly the effectives of wind-shear in converging ions into layers. The authors appear to be using winds averaged over an unspecified period to determine wind shear – yet there is no discussion of how valid this is, since sporadic E often have short lifetimes of only hours.

Response: Following your suggestion, we further discussed about the role of gravity waves in the formation of Es layers in the polar cap in the revised manuscript.

The wind shear mechanism does not work efficiently at either auroral zones or the magnetic equator (Haldoupis, 2012); therefore, the strong Es layers in the Earth's polar regions could be initially caused by gravity waves (Bautista et al., 1998; MacDougall et al., 2000a, b). In the simulations, the gravity waves with horizontal wavelengths smaller than ∼200 km are not explicitly resolved in WACCM (Liu et al, 2014). In particular, the vertical motion of gravity waves dominates the formation of Es layer in the polar cap, where the near-vertical magnetic field significantly reduces the effectives of wind-shear in converging ions into layers. Polar cap gravity waves were studied by

Johnson et al. (1995); MacDougall et al. (1997). These layers are maintained in an ionized state by charge exchange of neutral metal atoms with NO+ and O2+ ions by photoionization. These studies found that the vertical motion of gravity waves is very efficient in concentrating polar gap Es layers. The short-lived polar cap Es layers in winter appear to be associated with gravity waves. The polar cap Es layers in summer are long-lasting thin layers. These initial concentrations of metallic ions persist and change into long-lived Es because of ambient metallic ions. The Es layers at the cusp latitude are relatively different from those at the polar cap. The cusp Es could be associated with the convergence of ionization by the electric fields (MacDougall and Jayachandran, 2005).

Changes: Please see page 2 lines 2-4. "At high magnetic latitudes, the vertical motion of gravity waves is very efficient in concentrating ionization of Es becuase the magnetic field lines are nearly vertical in the polar gap (Bautista et al., 1998; MacDougall et al., 2000a, b; MacDougall and Jayachandran, 2005)."

Please see page 5 lines 2-5. "A difference between the Es intensity and EsOR distributions is at high latitudes: that is, the occurrence rates of Es are generally low (Arras et al., 2008), but the intensity of Es is relatively high. This pattern is more evident over the magnetic poles, which is likely the result of vertical motions of gravity waves in concentrating the ionization of Es layers (MacDougall et al., 2000a, b).

Please see page 7 lines 29-page 8 lines 13. "In our investigations, the global climatology of the intensity of Es layers is found to also have a seasonal dependence, with a pronounced maximum over midlatitudes in the summer hemisphere, as shown in Figure 5. . . .The cusp Es could be associated with convergence of ionization by the electric fields (MacDougall and Jayachandran, 2005)."

Another point is that the height distribution of sporadic E (Figure 1) shows a relatively large proportion of layers appearing between 40 and 90 km. This is dismissed here as an artefact, based on the fact that sporadic E should not form below 90 km because

the ion-neutral collision frequency is too high (page 3). The explanation for the artefact is unclear – reference is made to an "RO" event, though this term is not defined in the paper – but appears to be caused by slant viewing geometries. If that is the case, how reliable is the entire distribution of layers, including those above 90 km? The authors ought to show that the distribution they are using agrees with ionosonde measurements at a particular location.

Response: Thanks for your comments. In the revised manuscript, we included more references and explain the artefact more clearly. Most of the artefact are caused by integrated influence rather than slant viewing geometries. The Es layer over lower altitudes (between 40 and 90 km) should be some artefact resulting from mapping effect integrated along the LEO-GPS ray. As a result of the integrated influence either in the SNR or the slant TEC, the effect of Es layers in the high altitudes could map down to the lower tangent point altitudes, which may induce pseudo multiple peaks in one RO event (Zeng and Sokolovskiy, 2010; Yue et al., 2015).

We added a result (Figure 2) to show the entire distribution of the daily average Es intensity from 2006 to 2014 retrieved from COSMIC within $\pm 2.5°$ of latitude and longitude of one ionosonde station in Beijing ($40.3N°$, $116.2E°$), which agrees with ionosonde measurements.

Changes: Please see page 3 lines 30-page 4 lines 9. "Figure 1 shows the altitude distribution of the COSMIC S4max profiles. . . . Figure 2 shows the entire distribution of the daily average Es intensity from 2006 to 2014 retrieved from COSMIC within $\pm 2.5°$ of latitude and longitude of one ionosonde station in Beijing ($40.3N°$, $116.2E°$), which agrees with the ionosonde measurements (foEs)."

The paper ends on a vague and rather disappointing note: "It indicates that, in addition to the vertical windshear effects, other processes such as magnetic field effects, meteoric mass influx into the earth's atmosphere and chemical processes of metallic ions are also likely to play an dominant role in the geographical and seasonal variations of

Es layers." There is no attempt to explain how these "chemical processes", the meteoric mass influx, or unspecified "magnetic field effects" could explain the observations which do not accord with the wind shear theory.

Response: Thanks for your comments.

In the revised manuscript, we fulfilled the discussion section according to the reviewer's comments. We discussed more about the gravity waves and their dominant role in the polar gap Es layer (MacDougall et al., 2000a, b). The wind shear mechanism does not work efficiently at either auroral zones or the magnetic equator (Haldoupis, 2012); therefore, the Es layer in the North and South Poles are likely associated with polar gap gravity waves. The wind shear mechanism is inefficient at high geomagnetic latitudes because the magnetic field is nearly vertical. It is found that an initiation of Es layers caused by gravity wave vertical motion could account for the properties of poleward Es layers. The Es layers at the cusp latitude are relatively different from those in the polar cap. The cusp Es could be associated with convergence of ionization by the electric fields (MacDougall et al., 2005).

Many studies have investigated the global climatology of the Es layer occurrence rate (EsOR) but the global climatology of the intensity of Es layers has not be been fully studied. The purpose of the present paper is to study the global intensity of Es layer. Several new and interesting observations of Es intensity are presented. Our modest effort is to present our understanding of the layered phenomena of the Es layer as the topic is relevant in the special issue: layered phenomena in the mesopause region. Firstly, the high-latitude Es layers are quite intense in our study although their occurrence rate is low as reported in Chu et al. (2014); Shinagawa et al. (2017). Secondly, a noticeable global gap of Es is observed near the magnetic equator. Thirdly, clear features and geographical distribution with dependence of geomagnetic latitudes can be seen in the geomagnetic latitudes of 60-80° because of the increased spatial resolution.

As for the mechanism for the observations, our study shows the elemental mechanism responsible for Es layers based on the wind shear theory could explain the seasonal dependence of Es intensity (97-114 km) but it is hard to explain the Es seasonal dependence at higher altitudes. The wind shear theory is considered the primary theory to explain the physical production of Es layers (Whitehead, 1989; Haldoupis et al., 2007), although the overall morphology of Es has not be fully explained by wind shear effect until now, as shown in Figure 9 and Figure 10 in our study. In the simulation of wind shear theory explanation for the Es seasonal variation, we calculated the vertical ion convergence by using winds from a global chemistry-climate model (WACCM). In Section 6, we conclude the results of this study. It also notes that, to accurately understand the distribution of global metallic ions, we need to further investigate the geographical and seasonal variations in Es layers combined with the ground-based ionosonde observations. We are also trying to explain those phenomena in theory and another work of more comprehensive modelling are in progress and hope to be subsequently able to explain the phenomena. We hope our effort of this study as the relevant topic in the special issue in ACP could attract more attention and researches in mechanism of the Es formation, which is an ongoing problem in the ionosphere since 1961 (Whitehead 1961; Whitehead 1989).

Changes: Please see page 7 lines 29-page 8 lines 13. "In our investigations, the global climatology of the intensity of Es layers is found to also have a seasonal dependence, with a pronounced maximum over midlatitudes in the summer hemisphere, as shown in Figure 5. ...The cusp Es could be associated with convergence of ionization by the electric fields (MacDougall and Jayachandran, 2005)."

Please see page 9 lines 20-27. "Although the wind shear theory for the Es formation was conceived and formulated in 1960s (Whitehead, 1961), its importance for understanding the formation of Es must have escaped attention. This study implies that, in addition to the vertical wind shear effects, other processes, such as the vertical motion of gravity waves, magnetic field effects, meteoric mass influx into Earth's atmosphere

and the chemical processes of metallic ions, should also be considered, which could play a dominant role in the geographical and seasonal variations in Es layers. To accurately understand and properly quantify the properties of Es layers at a global scale that are also associated with the distribution of global metallic ions, we need to combine more ground-based ionosonde data with satellite observations and extensively study the geographical and seasonal variations in Es layers."

Specific points which need to be addressed: p. 4, line 10: why do you state "may be caused"? You have applied several different models, including WACCM. What do they tell you about the anomalies? The specified dynamics version of WACCM would be quite informative. You also give no details about the version of WACCM output that you use, etc.

Response: In addition to the seasonal variability of Es layers, the distribution of Es also has an interannual variability. In this paper, we focus on the seasonal variability of Es by using the WACCM wind to examine the wind shear effect. In Section 4, we find that, the wind shear theory could not fully explain the Es seasonal dependence in the simulation in view of the present findings. Figure 2 also shows a modification in Es latitudinal extension. Detailed investigations on this specific cause will be a topic of further studies.

Following your suggestion, we give more details about the version of WACCM output we use in the revised manuscript. Version 4 of the WACCM (WACCM4) is a global climate model with interactive chemistry, developed at the National Center for Atmospheric Research (NCAR) (Marsh et al., 2013). A specified dynamics run of WACCM4 (SD-WACCM4) was constrained by the Modern-Era Retrospective Analysis for Research and Applications (MERRA). SD-WACCM4 is used to simulate the global distribution of the divergence of ion velocity from the period of 2006 to 2014, which is consistent with the period of Es observations from the COSMIC occultation data.

Changes: Please see page 6 lines 11-15. "Version 4 of the WACCM (WACCM4) is

a global climate model with interactive chemistry, developed at the National Center for Atmospheric Research (NCAR) (Marsh et al., 2013). A specified dynamics run of WACCM4 (SD-WACCM4) was constrained by the Modern-Era Retrospective Analysis for Research and Applications (MERRA). SD-WACCM4 is used to simulate the global distribution of the divergence of ion velocity from the period of 2006 to 2014, which is consistent with the period of Es observations from the COSMIC occultation data."

p. 4, line 17: surely the wind shear mechanism should be most effective at the geomagnetic equator when the magnetic field is horizontal? Why do you state that a vertical component is required for ion convergence?

Response: Thanks for your comments. It is important to note that the wind shear mechanism does not work efficiently at either auroral zones or the magnetic equator (Haldoupis, 2012). The wind shear mechanism is inefficient at high geomagnetic latitudes because the magnetic field is nearly vertical. At the geomagnetic equator when the magnetic field is horizontal, in the zonal wind shear action, ions move vertically by Lorentz forcing. But they fail to converge into a layer because they are withheld by the magnetized electrons. The plasma maintains locally neutral. For a meridional wind shear process, ions move along the magnetic field lines with no Lorentz forces acted upon. Therefore, a noticeable gap near the magnetic equator is expected, explained by the vanishing vertical component of the geomagnetic field lines which keeps the ionized particles from effectively vertically converging. This gap could also be found in the distribution of EsOR though it is not as evident in Arras et al. (2008).

In the revised manuscript, we have explained it in more details.

Changes: Please see page 4 lines 25-31. "When the magnetic field is horizontal at the geomagnetic equator, under the zonal wind shear action, ions move vertically by Lorentz forcing. However, they fail to converge into a layer because they are withheld by magnetized electrons. The plasma remains locally neutral. For a meridional wind shear process, ions move along the magnetic field lines with no Lorentz forces acted upon

(Haldoupis, 2012). Therefore, a noticeable gap near the magnetic equator could be expected, which is explained by the vanishing vertical component of the geomagnetic field lines, keeping the ionized particles from effectively vertically converging. This gap could also be found in the distribution of EsOR although it is not as evident in Arras et al. (2008)."

p. 4, line 32: the sentence "The ionization of Es layers is persistently magnetic fields trapped in the polar regions" make no sense. Response: Thanks for your comments. The sentence was removed in the revised manuscript.

Changes: Done.

p. 5, line 18: the sentence "One of the unsolved issues in the ionosphere is that the well pronounced seasonal dependence of mid-latitude Es layers does not have a comprehensive explanation, which is inexplicable from the windshear theory" contradicts your own conclusion that wind shear does explain many of the mid-latitude features!

Response: Thanks for your comments. This sentence is changed as "one of the unsolved issues in the ionosphere is that the overall morphology including the seasonal dependence of Es layers does not have a comprehensive explanation (Whitehead, 1989; Haldoupis et al., 2007)."

The wind shear theory was formulated by Whitehead (1961). In his review article Whitehead (1989), it stated that the wind shear theory is the primary theory to explain Es layers. However, it does not explain the overall morphology of sporadic E, in particular the large summer maximum. That is the motivation of our simulation study in Section 4. In our simulations, we found the ion convergence could explain the seasonal dependence of mid-latitude Es layers at altitudes between 97 and 114 km but failed to explain at altitudes between 114 and 128 km. So we concluded that the convergence of vertical ion velocity could not fully but partially explain the seasonal dependence of Es intensity.

Changes: Please see page 5 lines 30-31. "One of the unsolved issues in the ionosphere is that the overall morphology, including the seasonal dependence of Es layers, does not have a comprehensive explanation (Whitehead, 1989; Haldoupis et al., 2007)."

p. 5, line 24: you suddenly mention Fe+ ions here. Why only Fe+, where do they come from? What is the evidence? Response: Sorry for the unclear statement. It is true that the equation (1) of ion velocity wi is a general expression, not a specific expression only for Fe+. In the revised manuscript, we changed this sentence.

Changes: Please see page 6 lines 3-4. "From the wind shear theory (e.g., Nygren et al., 1984; Mathews, 1998; Kirkwood and Nilsson, 2000), the vertical ion velocity wi induced by the neutral wind is described by equation (1):"

p. 6, line 1: why do you take the wind from WACCM, and the atmospheric composition from MSIS? This is inconsistent. This calculation should be performed using composition and winds from the same model. Response: Thanks for your comments. In this paper, to compare with previous studies, the neutral wind is provided by output from WACCM and the ion-neutral frequency is calculated by the atmospheric composition from MSIS-00 atmospheric model in accordance with Chu et al. (2014). Thus, the result both of simulations and Es intensity can be compared with the simulation and the Es layer occurrence rate (EsOR) results from Chu et al. (2014).

Besides, WACCM cannot directly provide atmospheric density estimates and its atmospheric density would also be calculated based on the ideal gas law.

We also compare the atmospheric density from MSIS-00 and WACCM. They have the similar spatial variation. The determining factor for the divergence of ion velocity variation is the wind field.

Changes: Please see page 6 lines 15-17. "To compare with previous studies, the neutral wind is provided by the output from WACCM, and the ion-neutral frequency is

calculated by the atmospheric composition from the MSIS-00 atmospheric model in accordance with Chu et al. (2014)."

p. 6, line 19: the windshear theory does not explain formation of Es layers at high geomagnetic latitudes, because the magnetic field is nearly vertical. There is evidence that within the polar cap gravity waves play a dominant role in Es formation (see, e.g. the papers by John MacDougall from Western Ontario). This is not discussed anywhere.

Response: Following your suggestion, we included the discussion of the role of the vertical motion of gravity waves in the formation of the polar cap Es layer (Bautista et al., 1998; MacDougall et al., 2000a, b; MacDougall and Jayachandran, 2005).

Changes: Please see page 2 lines 2-4. "At high magnetic latitudes, the vertical motion of gravity waves is very efficient in concentrating ionization of Es becuase the magnetic field lines are nearly vertical in the polar gap (Bautista et al., 1998; MacDougall et al., 2000a, b; MacDougall and Jayachandran, 2005)."

Please see page 5 lines 2-5. "A difference between the Es intensity and EsOR distributions is at high latitudes: that is, the occurrence rates of Es are generally low (Arras et al., 2008), but the intensity of Es is relatively high. This pattern is more evident over the magnetic poles, which is likely the result of vertical motions of gravity waves in concentrating the ionization of Es layers (MacDougall et al., 2000a, b)."

Please see page 7 lines 29-page 8 lines 13. "In our investigations, the global climatology of the intensity of Es layers is found to also have a seasonal dependence, with a pronounced maximum over midlatitudes in the summer hemisphere, as shown in Figure 5. ...The cusp Es could be associated with convergence of ionization by the electric fields (MacDougall and Jayachandran, 2005)."

p. 6, line 25: did you derive equation (3) here? If not, it should be referenced. Response: Yes, the equation (3) is correctly derived from the basic steady-state ion momentum equation, with the declination angle D into consideration.

[Figure]

p. 6, line 32: why are the regions of ion convergence different when the magnetic declination angle is included? You seem to imply that agreement with the observations is worse. What does that imply? Is equation (3) incorrect? Response: Thanks for your comments. The magnetic declination angle currently ranges from -30° to 26°; therefore its influence on the vertical ion velocity $w_i$ is expected. The expression of vertical ion velocity was often omitted and mathematically simplified with angle D=0, that is

To further investigate the magnetic field effects on the wind shear processes of Es formation, in this study, we take the magnetic declination angle D included in equation (3). The region with large value of D has a different result in the simulation which can be seen in Figure 10. The agreement with the observations becomes worse, which could imply that the cause of global Es layers remains a mystery because it cannot be fully accounted for by the wind shear effect (Whitehead 1989, Haldoupis 2007). The formation of mid-latitude Es layers could be partially explained by the wind shear theory. The investigation of causes of seasonal variability in Es should lead to more detailed studies to fully understand and properly quantify the properties of Es layers.

The equation (3) is correct, which is derived from the basic steady-state ion momentum equation.

Changes: Please see page 7 lines 18-21. "The agreement with the observations becomes worse, which could imply that the cause of global Es layers remains a mystery because it cannot be fully accounted for by the wind shear effect (Whitehead, 1989; Haldoupis et al., 2007). The formation of mid-latitude Es layers could be partially explained by the wind shear theory. The investigation of causes of seasonal variability in Es should lead to more detailed studies to fully understand and properly quantify the properties of Es layers."

p. 7, line 11: why would energetic particle precipitation and cosmic rays generate Es in the polar region? Most cosmic ray ionization occurs around the tropopause, and EPP would not create a thin layer of ions. Response: Thanks for your comments. In this

revised manuscript, we remove this sentence and include the role of the vertical motion of gravity waves in the formation of Es layers in the polar cap.

Changes: Please see page 8 lines 2-4. "The wind shear mechanism does not work efficiently at either auroral zones or the magnetic equator (Haldoupis 2012); therefore, the strong Es layers in the Earth's polar regions could be initially caused by gravity waves (Bautista et al., 1998; MacDougall et al., 2000a, b)."

p. 8, line 2: I thought this range of IDP input had been considerably reduced in the Carrillo-Sanchez et al., GRL 2016 paper (which followed the one you cite). What difference would it make anyway to sporadic E formation? Response: Thanks for your comments. We have read and cited this paper. Carrillo-Sanchez et al. (2016) employed a different approach to get an estimate of the total input mass (43±14 tons per day), but the estimates of the global Interplanetary Dust Particles (IDPs) are different, depending on the methods used to make the estimate (Carrillo-Sanchez et al., 2016; Plane, 2012). Carrillo-Sanchez et al. (2016) estimated the IDP from three observations to constrain the relative contributions of each dust source, with lidar observations of the vertical Na and Fe fluxes.

The behaviour and climatology of the Es layer are related to the distribution of meteoric smoke particles (MSPs) deposition, which provide a permanent sink for gas-phase metallic compounds. The global IDP input could influence the total MSP volume densities to the earth's surface. If the upper range of estimates is correct, then the vertical transport should be considerably faster than is generally thought to be the case, so that metallic ions (the Es layer) transports vertically and is removed more rapidly in order to sustain a higher rate of injection; vice versa (Plane, 2012). Besides, to estimate the meteoric mass influx accurately, as one of the processes determining the Es formation, could help precisely study the geographical and seasonal variations in Es layers.

Changes: Please see page 9 lines 2-7. "The global input of meteoric material is well established to enhance the mesospheric metal layers and Es layers (Plane, 2004;

[Figure]

Carrillo-Sánchez et al., 2015; Plane et al., 2015), but the daily amount is still not well defined, and estimates of the global interplanetary dust particles (IDPs) range from 5–270 tonnes per day (Plane, 2012; Carrillo-Sánchez et al., 2016). These effects of meteoric ablation are significantly influenced by the magnitude of the IDP input by two orders of magnitude uncertainty. On the other hand, this fact also highlights the importance of fundamental understanding of the global climatology of Es layers."

Corrections: Need to indicate in the figure captions for Figure 3, 4, 5, 6, 7, 8 and 9 whether this is data averaged over the whole period or for a single year.

References: many do not have the complete initials of the authors

p. 1, line 3: "show a high . . . distribution and . . ."

p.1, line 6: " . . .bands. Simulations results show that . . ."

p. 1, line 12: ". . .layers are thin-layered . . ."

p. 1, line 19: "equatorial region"

p. 2, line 1: ". . .irregularities and their sharp . . .."

p. 2, line 33: ". . . models, namely the Whole . . ."

p. 3, line 2: ". . . time. Section 5. . ."

p. 3, line 6: ". . .behind earth's . . ."

p. 3, line 7: ". . .signal is received at . . ."

p. 3, line 16: ". . .time series . . ."

p. 6, line 9: ". . . seasons. They also showed . . ."

p. 7, line 31: ". . .provides a much greater . . ."

There are many other grammatical errors in the paper which need to be corrected.

Response: Thanks. We have corrected it.

Reviewer #3 comments (RC2): This paper reports some novel data and modelling on the occurrence and intensity of global Sporadic E layers and provides some interesting perspectives on the formation mechanisms of these layers. The data are derived from measurements of the S4 index from radio occultation measurements made by the COSMIC satellite constellation and the global distribution of Sporadic E which they reveal is similar to those derived from previous studies, with a strong occurrence peak in the mid-latitudes of the summer hemisphere. The authors comment on some interesting distinctions between occurrence and intensity of sporadic E layers; for example they notes that high-latitude layers, while being lower occurrence, tend to be quite intense when they do arise.

Response: Thank you for your positive comments.

Many papers have reported the global climatology of the Es layer occurrence rate (EsOR) by using satellite GPS RO measurements (Wu et al., 2005; Arras et al., 2008; Chu et al., 2014; Shinagawa et al., 2017), but the global climatology of the intensity of the Es layers has not be been fully studied. The purpose of the present paper is to study the global intensity of Es layers and compare the results of Es intensity with previous studies on the EsOR.

These are interesting data sets, but are somewhat spoiled by their relatively poor presentation. For example it would be nice to have seen graphs of global Sporadic-E occurrence and global Sporadic-E intensity in the same format, whereas what we actually see is a map of global occurrence (presumably averaged across seasons) in Figure 3 and then separate maps of intensity for each season in Figure 4.

Response: Sorry for the misleading captions of Figure 4 and Figure 5 (formerly Figure 3 and Figure 4). Figure 4 is the global geographical distributions of the Es average intensity from 2006-2014, and Figure 5 is the Es average intensity from 2006-2014 for four different seasons. We have revised the captions of Figure 4 and Figure 5 and

made it clearer.

Changes: Please see page 17 and page 18.

The most interesting part of the study is the attempt to explain the occurrence and intensity of the Sporadic E layers in terms of modelled neutral wind convergence, using wind fields from the WACCM model. This provides qualitative agreement with the observations, if it is assumed that the layers are due to neutral wind convergences in the lower E-region (97-114 km) but strongly suggests that wind convergences at higher E-region altitudes cannot explain the observations. A nice feature of the wind field modelling is that the magnetic declination has been properly included in the calculations and it is demonstrated that allowing for this factor changes the expected distribution of the modelled wind convergences significantly.

Response: Thank you for your positive comments. Our study focuses on the global climatology of the intensity of Es layers, which has not been fully studied. Several new observations of Es intensity are presented. Then, the global intensity of Es layers can be compared with previous studies on the EsOR.

Unfortunately, however, the paper is not able to make any firm conclusions, because the correspondence between the modelled wind convergences and the occurrences and intensities of the Sporadic E layers remains only qualitative at best. This almost certainly illustrates the deficiencies of the modelling assumptions. The wind fields, for example, are obviously idealised and must have significantly greater variability than the modelling suggests, an idea reinforced by the inter-annual changes in the occurrence data shown in Figure 2. In addition the authors comment on various other factors such as the variability of the meteor flux, the effects of geomagnetic storms and the effect of meteorological processes, any and all of which could result in differences between the modelling and the observations, but which would be hard to account for without much more complicated modelling. As a result, the interesting features which are observed are not very well explained.

Response: Thanks for your comments. The focus of the present paper is to study the global climatology of the intensity of Es layers. By comparing the occurrence rate and intensity of Es layers, several new and interesting observations of Es intensity are presented. Firstly, the high-latitude Es layers are quite intense in our study although the Es occurrence rate is low as reported in Chu et al. (2014); Shinagawa et al. (2017). Secondly, a noticeable global gap of Es is observed near the magnetic equator. Thirdly, clear features and geographical distribution with dependence of geomagnetic latitudes can be seen in the geomagnetic latitude of 60-80° because of the increased spatial resolution.

In Section 4, we simulated the global mean divergence of the vertical ion velocity using WACCM wind field from 2006 to 2014. The simulation result in Figure 8 shows that the elemental mechanism responsible for Es layers based on the wind shear theory could explain the seasonal dependence of Es intensity (97-114 km). It is also consistent with Chu et al. (2014).

However, more attention should be paid to the results of Figure 9 and Figure 10: that is some disagreements between simulations and results. The simulation in Figure 9 indicates that the wind shear theory alone has difficulty explaining the Es seasonal dependence at higher altitudes (114-128 km), although the wind shear theory is considered the primary theory to explain the physical production of Es layers (Whitehead, 1989; Haldoupis et al., 2007). Thus, the magnetic declination has been properly included in the expression of vertical ion velocity and in the simulation, on the basis of the steady-state ion momentum equation. The agreement with observations becomes worse in Figure 10, which could imply that the cause of global Es layers remains a mystery and should lead to more detailed studies to identify and quantify the formation of Es layers. Please note that the wind shear theory was conceived and formulated by Whitehead (1961). It was stated in Whitehead (1989) that "We conclude that the wind shear theory is the only viable theory that explains the detailed production of the layers. Nevertheless, it does not explain the overall morphology of sporadic E, in particular the

large summer maximum."

I agree that more complicated modelling is needed to well explain the interesting observations of the Es intensity. Our purpose here is to report the new observations of the global climatology of the Es intensity and try to explain the Es seasonal dependence by the wind shear theory in the simulation.

We are also trying to explain those phenomena, considering more other factors in theory. Another work of more comprehensive modelling is in progress and hope to be subsequently able to quantitatively explain the phenomena and disagreements between observations and simulations from the wind shear theory. We hope this paper could attract more attention to the ongoing mystery of the Es seasonal dependence. It is desirable to combine more ground-based ionosonde data with satellite observations, and study the spatial Es variations and regional features in more details

Changes: Please see page 9 lines 9-27. "In this study, we investigate the long-term climatology of the intensity of Es layers on the basis of S4max data retrieved from COSMIC GPS RO measurements. . . .Although the wind shear theory for the Es formation was conceived and formulated in 1960s (Whitehead, 1961), its importance for understanding the formation of Es must have escaped attention. . . . To accurately understand and properly quantify the properties of Es layers at a global scale that are also associated with the distribution of global metallic ions, we need to combine more ground-based ionosonde data with satellite observations and extensively study the geographical and seasonal variations in Es layers.

The language of the paper could be improved. It is occasionally imprecise, so that the meaning can be hard to decrypt. There are also some mistakes in spelling and grammar. These are not really what weaken the paper, however. The fundamental problem is that the processes which produce the Sporadic E layers are likely to have such a complex variability that no simple model can do a good job of characterizing them, and this is what the study ultimately shows.

Response: Thanks for your comments. We apologize for the mistakes in spelling and grammar. We have corrected it. As non-native English speakers, we have tried our best to improve the language of the paper and undertaken a further proof-reading and update of the manuscript. We present our study of the global climatology of the intensity of Es layers. We report some new observations of the global climatology of the intensity of Es layers, which have several different features from the occurrence rate of Es layers in Chu et al. (2014); Shinagawa et al. (2017). The simulation results in our study shows that the convergence of the vertical ion velocity could partially explain the seasonal dependence of the Es intensity while some disagreements between simulations and observations should be paid more attention. Although the wind shear theory for the Es formation was conceived and formulated in 1960s (Whitehead 1961), its importance for understanding the formation of Es must have escaped attention. We hope our modest effort of this study as the relevant topic in the special issue in ACP: layered phenomena in the mesopause region could attract more attention and researches in mechanism of Es formation, to better understand and properly quantify the properties of Es layers in the mesosphere and lower thermosphere region.
* * *
[Figure]

[Figure]

**Fig. 1.** Figure 2. The entire distribution of the daily average Es intensity retrieved from COSMIC within ±2.5° of latitude and longitude of one ionosonde station and ionosonde data (foEs) in Beijing from 2006